# ST-TGR: Spatio-Temporal Representation Learning for Skeleton-Based Teaching Gesture Recognition

**DOI:** 10.3390/s24082589

**Published:** 2024-04-18

**Authors:** Zengzhao Chen, Wenkai Huang, Hai Liu, Zhuo Wang, Yuqun Wen, Shengming Wang

**Affiliations:** 1Faculty of Artificial Intelligence in Education, Central China Normal University, Wuhan 430079, China; llczz@163.com (Z.C.); wkhuang@mails.ccnu.edu.cn (W.H.); hailiu0204@ccnu.edu.cn (H.L.);; 2National Engineering Research Center for E-Learning, Central China Normal University, Wuhan 430079, China; 3Faculty of Literature and Journalism, Xiangtan University, Xiangtan 411105, China; 4National Engineering Research Center of Big Data, Center China Normal University, Wuhan 430079, China

**Keywords:** classroom scenario, teaching gesture, pose estimation, action recognition

## Abstract

Teaching gesture recognition is a technique used to recognize the hand movements of teachers in classroom teaching scenarios. This technology is widely used in education, including for classroom teaching evaluation, enhancing online teaching, and assisting special education. However, current research on gesture recognition in teaching mainly focuses on detecting the static gestures of individual students and analyzing their classroom behavior. To analyze the teacher’s gestures and mitigate the difficulty of single-target dynamic gesture recognition in multi-person teaching scenarios, this paper proposes skeleton-based teaching gesture recognition (ST-TGR), which learns through spatio-temporal representation. This method mainly uses the human pose estimation technique RTMPose to extract the coordinates of the keypoints of the teacher’s skeleton and then inputs the recognized sequence of the teacher’s skeleton into the MoGRU action recognition network for classifying gesture actions. The MoGRU action recognition module mainly learns the spatio-temporal representation of target actions by stacking a multi-scale bidirectional gated recurrent unit (BiGRU) and using improved attention mechanism modules. To validate the generalization of the action recognition network model, we conducted comparative experiments on datasets including NTU RGB+D 60, UT-Kinect Action3D, SBU Kinect Interaction, and Florence 3D. The results indicate that, compared with most existing baseline models, the model proposed in this article exhibits better performance in recognition accuracy and speed.

## 1. Introduction

Teaching gestures, as a non-verbal teaching behavior, play an important role in the classroom [1], and gesture recognition technology can promote the application of teaching gestures in education and teaching [2]. For example, teaching gesture recognition can be used for classroom teaching evaluation, which can then assist teacher training. By analyzing the teaching gestures of excellent teachers, feedback and suggestions can be provided to other young teachers, helping them improve their teaching methods [3]. At the same time, gesture recognition can help create a richer and more interactive learning experience in online teaching environments [4]. For special education, such as students with hearing impairments, gesture recognition can also be used to identify and learn sign language [5,6]. Teaching gesture recognition brings many possibilities to education by improving teaching quality and enhancing student learning experience.

In recent years, the gesture recognition task, a noteworthy and highly challenging research topic in computer vision, has ignited the attention and discussion of many researchers [7]. How to extend the current gesture recognition technology to classroom teaching has aroused in-depth research in academia and industry. Specifically, gesture recognition leveraging computer vision can be categorized into two types: static gesture recognition and dynamic gesture recognition. The subject of static gesture recognition is the gesture image at a certain moment, and its recognition results are closely related to the appearance features of the hand in the image, such as contour, position, and texture [8]. The subject of dynamic gesture recognition is a sequence of images over a continuous period, and its recognition results are related to the appearance characteristics of the hand in the image, as well as the time series characteristics that depict the hand’s motion trajectory in the sequence [9]. In comparison with static gestures, dynamic gestures have richer diversity, practicality, and expressiveness. For the research on gesture recognition for classroom teachers, utilizing dynamic gesture recognition methods is also more effective.

Early traditional research on gesture recognition primarily focused on static gesture actions collected at close range. This involved first extracting the hand area of the target from the video frame image, then using manually extracted features and feature encoding methods to obtain hand region information in images [10], and finally classifying gesture actions through hand feature information [11]. The most common gesture detection methods incorporate hand-based skin color, shape, pixel values, 3D models, and motion features. However, these methods are susceptible to changes in lighting, skin color differences, background interference, natural changes, and self-occlusion of fingers, as shown in Figure 1, resulting in unsatisfactory detection results and slow computation speeds, introducing challenges to meeting real-time requirements in practical applications. With the advancement of deep learning, hand pose detectors based on deep learning are gradually gaining a mainstream position [12]. In comparison with manual feature extraction detection, this method exhibits better recognition efficiency and generalization ability. However, the current application of gesture recognition research in classroom teaching primarily focuses on the student level, inferring their classroom behavior by integrating their static posture information and overlooking the potential impact of teaching gesture actions on classroom teaching behavior [13].

The main contributions of this paper are as follows:(i)To mitigate the difficulty of single-target dynamic gesture recognition in multi-person scenarios, we propose a gesture recognition algorithm based on skeleton keypoints. Our method mainly extracts the skeleton keypoint coordinates of the target through human pose estimation technology and then inputs information sequences of different scales into subsequent gesture recognition modules for gesture action classification.(ii)A simple and efficient action recognition network module MoGRU is proposed in this paper, which integrates multi-scale bidirectional GRU modules and improved attention mechanism modules. It can achieve good action classification performance on different benchmark action datasets when only using target skeletal information, especially when dealing with small sample datasets. In addition, this module has a good balance between recognition speed and recognition accuracy, bringing possibilities for practical applications.(iii)To promote the application of gesture recognition in teaching, this article constructs a teaching gesture action dataset (TGAD) based on a real classroom teaching scenario, which includes four types of teaching gesture actions from different perspectives, totaling 400 samples. After model testing, our proposed method can achieve 93.5% recognition accuracy on this dataset.

The remaining part of this article is structured as follows: Section 2 reviews the relevant literature. Section 3 introduces the teaching gesture action recognition algorithm based on the skeletal keypoints proposed in this paper. Section 4 displays the experimental process and analysis results of the algorithm. Section 5 summarizes this article.

## 2. Related Work

In this section, we reviewed the relevant work on methods involved in gesture recognition based on skeletal keypoints. The main content encompasses three parts: skeleton-based action recognition, 2D multi-person pose estimation, and the incorporation of the attention mechanism.

### 2.1. Skeleton-Based Action Recognition

The main task of action recognition is to recognize human behavior and behaviors in videos. Action recognition methods based on deep learning can be divided into two categories: skeleton-based and video-based, depending on whether human keypoints are initially detected. This section will focus on current action recognition algorithms that are related to the skeleton-based category. At present, deep learning-based methods can be roughly divided into three subcategories according to the different network structures of the model: recurrent neural networks (RNNs), convolutional neural networks (CNNs), and graph convolutional networks (GCNs).

RNNs have temporal memory capabilities, making them particularly effective in processing temporal data with long-term dependencies. To address the difficulty of classifying fine-grained behaviors using a single network model, Li et al. [14] proposed an adaptive RNN tree model. Gao et al. [15] proposed a variable speed IndRNN model, which adaptively adjusts the learning rate to make the network more robust to different sampling rates and execution speeds of different sequences. Ryumin et al. [16] proposed using spatio-temporal feature fusion combined with bidirectional LSTM modules to construct end-to-end network models. However, RNN-based models often overly emphasize the temporal information of actions, and the modeling effect on spatial information is not ideal.

Compared with RNNs, CNNs have strong information extraction capabilities and can efficiently perform spatial modeling to learn semantic information. Tu et al. [17] proposed a dual-stream 3D CNN that uses convolutional kernels of different scales to capture large-scale temporal information and transform bone data into multi-temporal sequences. A fusion CNN model was proposed by Li et al. [18], which encodes the spatio-temporal information of bone data into skeleton trajectory shape images (STSIs) and skeleton pose images (SPIs) through grayscale values. Although this type of method utilizes CNN modules with strong spatial modeling capabilities, its “local connectivity” property ignores the expression of semantics between distant joint points, and the generation of encoding graphs is complex, which is not ideal for optimization and improvement.

The human skeleton is a naturally occurring topological structure. Unlike RNNs and CNNs, GCNs are more suitable for processing non-Euclidean data [19]. The main contribution of the spatio-temporal GCN (ST-GCN) network proposed by Yan et al. [20] is the use of multi-layer graph convolution to extract spatio-temporal features of bones and construct spatio-temporal maps. It represents the physical structure of the human body through joints and spatial edges, adds time edges to replace the original complex optical flow, and simplifies weight assignments based on partitioning strategies. A channel-wise topology refinement GCN was proposed by Chen et al. [21] to dynamically learn topology and aggregate joint features in different channels. Chi et al. [22] used the self-attention based graph convolution module to infer the contextual intrinsic topology of bone information in spatial modeling. A novel graph convolutional network module and separable temporal convolutional network (TCN) for extracting sign language information were proposed by Jiang et al. [23] for multi-modal gesture recognition. Although the GCN model is more suitable for handling human topological structures and has shown better performance than CNN and RNN in bone-based action recognition in recent years, its graph structure adaptation ability is not strong because stacking too many layers in the network can lead to a decrease in its computational performance.

### 2.2. 2D Multi-Person Pose Estimation

Pose estimation involves estimating the position of keypoints in the human body, such as the head, hands, and body [24]. It is the foundation of many high-level semantic tasks, such as action recognition and abnormal behavior detection [25]. Based on the application scenarios proposed in this article, this section will concentrate on discussing the research overview of the existing literature on 2D multi-person pose estimation problems. First, 2D multi-person pose estimation methods based on deep learning can be generally divided into two categories.

One is the “top-down” method, which typically uses a human object detector to obtain a set of bounding boxes from the input image and then directly uses existing single-person pose estimators to predict the person’s pose [26]. Given that the posture predicted by this method heavily relies on the accuracy of object detection, most current research primarily focuses on optimizing existing human object detectors, such as faster R-CNN, feature pyramid network, and other network structures. Fang et al. [27] used a spatial transformer network, non-maximum suppression, and an hourglass module to improve pose estimation accuracy. Xiao et al. [28] incorporated several deconvolution layers into the last convolutional layer of ResNet to generate more accurate heatmaps based on deep and low-resolution features. High-Resolution Network (HRNet), which was proposed by Wang et al. [29], is used to exchange high- and low-resolution representation information, thereby maintaining high-resolution representation information during processing. However, the processing speed of this model algorithm is limited by the amount of detection personnel required for the image.

Another method adopts a “bottom-up” strategy, which directly predicts all joints of each individual and then assembles them into independent human skeletons. This method mainly consists of two parts: the detection of human joints and candidate joint grouping. Among them, Cao et al. [30] proposed using a convolutional pose machine method to predict all human joints with partial affinity fields. The pose partition network proposed by Nie et al. [31] can be used for joint detection and dense regression to achieve the segmentation of joints. Kreiss et al. [32] constructed a PifPaf network that enhances heatmap accuracy at high resolutions through a part intensity field (“Pif”) and connects body joints using a part association field (“Paf”). This type of “bottom-up” method can achieve a faster detection speed compared with most “top-down” methods, but the correct combination of joint points in complex environments will be a challenging research task.

### 2.3. Attention Mechanism

Attention mechanism is a technique used to simulate human visual processing of complex information, which can be applied to various deep learning models in different fields and tasks [33]. With the recent widespread application of attention mechanisms in computer vision, many researchers have attempted to integrate this mechanism into action recognition, using different forms of attention mechanism modules, such as self-attention, multi-head self-attention, and spatio-temporal self-attention, etc., which can bring different performances.

Rohit Girdhar et al. [34] proposed attention pooling to replace the commonly used mean pooling or max pooling in the final pooling layer of CNN network structures and constrained the attention through human pose keypoints to better converge to the corresponding action category in the final network. The DSTANet network proposed by Lei et al. [35] allows modeling of spatio-temporal dependencies between joints by constructing attention blocks, without the need to know their positions or interconnections. Specifically, three techniques for constructing attention blocks have been proposed in this paper to meet the specific requirements of bone data: spatio-temporal attention decoupling, decoupled position encoding, and spatial global regularization. Wang et al. [36] proposed a plug-and-play hybrid attention mechanism called the ACTION module for temporal action recognition (such as gestures). This module mainly includes spatio-temporal attention, channel attention, and motion attention, resulting in better action recognition results. Although using attention mechanisms with different characteristics can improve the performance of deep learning models, how to select and improve an effective attention module has become a difficult point in current research.

## 3. Methods

### 3.1. Overview of ST-TGR Model

To optimize and mitigate the difficulty of dynamic gesture recognition for teachers in real teaching scenarios, this article proposes a teaching gesture action recognition algorithm based on skeleton keypoints. This algorithm mainly consists of two network modules. The first module uses a high-performance human pose estimation detector RTMPose [37] based on the MMPose algorithm library to recognize teacher skeletal keypoints in classroom teaching videos. The second module uses a preset sliding window to feed keypoint sequences of skeletons of different scales to the subsequently constructed action recognition network MoGRU for classification and localization. By combining these two modules, teaching gesture action recognition can achieve fast and accurate results. The overall architecture of the network model is shown in Figure 2.

### 3.2. Skeleton Keypoint Extraction

Although many studies have achieved good results in 2D multi-person pose estimation, in practical application scenarios, challenges remain such as complex model computation and large parameter quantities, leading to high calculation delays. To enhance the performance of multi-person pose estimation in real teaching scenarios, this paper adopts the high-performance human pose estimation detection technology RTMPose based on the MMPose algorithm library, which is a model that can achieve accurate real-time detection in practical application scenarios.

In terms of recognition accuracy, RTMPose follows the “top-down” pattern, which uses ready-made detectors to obtain bounding boxes and then estimates each person’s pose separately. This method has a more accurate recognition effect compared with “bottom-up” algorithms. When facing complex classroom teaching environments with multiple people, inevitable occlusion issues arise between teachers, students, and the environment. Using a precise and effective posture detector can bring better performance for subsequent action recognition. Further, RTMPose adopts a SimCC-based algorithm for keypoint prediction, which considers keypoint localization as a classification task. Compared with heatmap-based algorithms, SimCC-based algorithms maintain competitive accuracy while reducing computational workload. In Section 5 of this article, we also validated this viewpoint through comparative experiments.

In terms of recognition speed, RTMPose adopts CSPNeXt, which was originally designed to cope with object detection tasks, as the backbone structure. Although this backbone structure is not the optimal choice for intensive prediction tasks such as pose estimation and semantic segmentation, CSPNeXt can achieve a better balance between speed and accuracy, and it is also easy to deploy in subsequent models. To improve the inference speed of the network module, RTMPose uses the skip frame detection strategy proposed in Blazepose [38] to accelerate inference speed, as shown in Figure 3, and improves pose processing through non-maximum suppression and smoothing filtering, thereby achieving better robustness.

In the training process of the model, to further utilize global and local spatial information, drawing inspiration from Tokenpose [39], we used a self-attention module to refine the keypoint representation, and we adopted a transformer variant, namely, the gated attention unit (GAU). Compared with regular transformers, the GAU has a faster speed, lower memory cost, and better performance. The GAU improves the feed-forward network in the transformer layer using Gated Linear Units and elegantly integrates attention mechanisms. Equation (Equation 1) is shown as follows, where ⊙ is element-wise multiplication, and ϕ is the activation function.
(1)$$\left\{\begin{array}{cc} U & =\phi_{u}\left(XW_{u}\right),\\ V & =\phi_{v}\left(XW_{v}\right),\\ O & =\left(U⊙AV\right)W_{o}. \end{array}\right.$$

The calculation of the attention (*A*) module is shown in Equation (Equation 2), where *S* is 128 and *Q* and *K* are the results of linear variation.
(2)$$\begin{array}{c} A=\frac{1}{n}{\mathrm{Relu}}^{2}\left(\frac{Q\left(X\right)K{\left(Z\right)}^{⊤}}{\sqrt{s}}\right),Z=\phi_{z}\left(XW_{z}\right). \end{array}$$

When calculating the training loss, the SimCC-based approach is adopted to treat coordinate classification as an ordered regression task, and the soft label encoding proposed in SORD is followed. Meanwhile, in the Softmax operation, temperature is added to the model output and soft labels to further adjust the normalized distribution shape, as shown in Equation (Equation 3). Among them, $\phi_{\left(r_{t},r_{i}\right)}$ is the selected metric loss function, which punishes the $r_{i\in Y}$ distance between the true metric value of $r_{t}$ and the rank. At the same time, a non-normalized Gaussian distribution is used as the inter-class distance measure, and the calculation method is shown in Equation (4): (3)$$\begin{array}{cc} y_{i} & =\frac{e^{-\phi (r_{t},r_{i})/\tau }}{\sum_{l=1}^{L}e^{-\phi (r_{t},r_{l})/\tau }}, \end{array}$$(4)$$\begin{array}{cc} \; & \phi \left(r_{t},r_{i}\right)=e^{\frac{-{\left(r_{t}-r_{i}\right)}^{2}}{2\sigma^{2}}}. \end{array}$$

Finally, using the RTMPose algorithm, we convert the original input video file into a skeleton sequence corresponding to the frame. The dimension of the output vector is the matrix $x\in R^{L\times K}$, where *L* is the length of the sequence in time steps and *K* is the product of the number of joint points and the dimension. For example, considering 2D single-person pose estimation based on the COCO dataset [40], where each image contains 17 pieces of joint information of a target object, and each piece of joint information has *x* and *y* coordinates, the value of *K* is 2 × 17 = 34. In order to cover different application scenarios, the algorithm library provides a series of model configuration files with different parameter sizes. After a series of control experiments, this article selected the RTMPose-m model as the preliminary teacher skeletal keypoint detector and deployed the TensorRT inference framework in subsequent practical applications to accelerate the inference process.

### 3.3. Classification of Gesture Actions

Given the strong dependence of gesture recognition tasks on temporal and spatial information, previous algorithm models that only used RNN or CNN did not achieve optimal recognition results. Therefore, to fully consider how to better integrate these two types of information, this article proposes a new MoGRU action recognition network model which includes three layers of bidirectional GRU modules, multi-layer CNN modules, and an improved multi-head self-attention module. Experimental verification shows that this model has an excellent ability to extract spatio-temporal feature information and can effectively process the sequence information of keypoints in teacher bones.

In terms of the temporal information dimension, considering that the algorithm model constructed based on recurrent neural networks can exhibit higher sensitivity to the temporal information of sequences, this paper selects a bidirectional gated recurrent unit (BiGRU), shown in Figure 4, as the main structure of the model after comparing the computational complexity and recognition accuracy of various recurrent neural network modules. Compared with traditional recurrent neural networks, a GRU can better capture long-range dependencies of long sequences and effectively alleviate the problem of gradient vanishing. Compared with LSTM, a GRU has fewer parameters, simpler calculations, and a faster training speed. Specifically, a GRU has only two gate structures (different from LSTM’s input gate, forget gate, and output gate), one reset gate *r* and one update gate *z*. The reset gate determines whether to ignore past state information, while the update gate determines the proportion allocated between the previous state information and the new information at the current time. The calculation Equation (Equation 5) for its model is as follows:(5)$$\left\{\begin{array}{cc} \; & r_{t}=\sigma \left(W_{r}x_{t}+U_{r}h_{t-1}+b_{r}\right),\\ \; & z_{t}=\sigma \left(W_{z}x_{t}+U_{z}h_{t-1}+b_{z}\right),\\ \; & \tilde{h_{t}}=\tanh\left(Wx_{t}+U\left(r_{t}⊙h_{t-1}\right)+b\right),\\ \; & h_{t}=\left(1-z_{t}\right)⊙h_{t-1}+z_{t}\tilde{h_{t}}. \end{array}\right.$$

Here, *t* represents the current time step, *x* represents the input, and *h* represents the hidden state. Among them, the hidden state vector dimensions of the three-layer GRU used in this article are 256, 512, and 128, respectively. Through the conversion of multi-layer GRU modules, we can transform the frame-related keypoints information formed by the original prediction into a feature vector $c\in R^{L\times 128}$ containing temporal information of teacher gesture actions. At the same time, by adopting a bidirectional connection strategy, the input skeleton sequence information can be better utilized, thereby increasing the accuracy of the model for gesture classification. The dimension of the feature vector output by this process is $c^{\prime }\in R^{L\times 256}$.

In terms of the spatial information dimension, after obtaining the temporal feature vectors $c^{\prime }$ of skeletal keypoints, we utilized convolutional neural network modules of different scales to extract spatial information between keypoints at the same frame time, allowing the model to better understand the correlation between keypoints. To preserve the dimension of the feature vectors during the convolution operation, we pad the feature vectors to a certain extent and use 1 × 1 convolution kernels to achieve the fusion of feature information. The calculation Equation (Equation 6) is as follows, where *K* is the size of the convolution kernel, $N_{i}$ is the size of the output vector feature, $C_{in}$ is the channel dimension of the input vector feature, and $C_{out_{j}}$ is the channel dimension of the output vector feature.
(6)$$out\left(N_{i},C_{out_{j}}\right)=bias\left(C_{out_{j}}\right)+\sum_{k=0}^{C_{in-1}}weight(C_{out_{j}},k)\times input(N_{i},k)$$

To further enhance the correlation between joint points in different frame time sequences, we employed an improved multi-head self-attention mechanism module after the convolution module to enhance spatio-temporal information features. Specifically, we add the vector features calculated by the convolution module to the multi-head self-attention mechanism module, and generate the corresponding Query, Key, Value using a linear change in the context vector dimension of 256. For each attention head in the module, we calculate the attention weight to balance the degree of correlation in the input information. Among them, the calculation method of the attention weight adopts the dot product operation, and the calculated attention score is normalized through the softmax function. The calculation Equation (Equation 7) for the attention mechanism module is as follows:(7)$$\left\{\begin{array}{cc} \; & Similarity\left(Query,Key_{i}\right)=Query\cdot Key_{i},\\ \; & a_{i}=Softmax\left(Sim_{i}\right)=\frac{e^{Sim_{i}}}{\Sigma_{j=1}^{L\chi }e^{Sim_{j}}},\\ \; & Attention(Query,Source)=\sum_{i=1}^{L_{x}}a_{i}\cdot Value_{i}. \end{array}\right.$$

Finally, we input the fused spatio-temporal information feature vectors into the fully connected layer for softmax classification prediction. The cross-entropy loss function is used to calculate loss during the training process. Equations (Equation 8) and (9) for this process are as follows, where *a* is the calculation result of softmax and *y* is the label of the training sample. The encoding method of the label adopts the one-hot encoding format.
(8)$$\begin{array}{cc} \; & softmax\left(x_{i}\right)=\frac{e^{x_{i}}}{\sum_{k}e^{x_{k}}}, \end{array}$$
(9)$$\begin{array}{cc} \; & L=Loss\left(a,y\right)=-\sum_{j}y_{j}\ln a_{j}. \end{array}$$

## 4. Experimental Results and Analysis

### 4.1. Dataset

NTU RGB+D [41]: This dataset is a large-scale dataset for RGB-D human action recognition. It contains 60 types of actions, with a total of 56,880 samples, of which 40 are daily behavioral actions, 9 are health-related actions, and 11 are mutual actions between two people. These actions were completed by 40 people aged from 10 to 35 years old. This dataset was collected by Microsoft Kinect v2 sensors, and three cameras were used from different angles. The vertical heights of the three cameras were the same, with horizontal angles of −45°, 0°, and +45°, respectively. The collected data include depth information, 3D skeleton information, RGB frames, and infrared sequences.

SBU Kinect Interaction [42]: This dataset is an action recognition dataset captured by Kinect cameras and primarily describes the interaction behavior of two people. All the videos were recorded in an identical laboratory environment. Seven participants engage in pairwise interaction, and in most activities, one person makes an action, while the other reacts. Each action category contains either one or two sequences. The entire dataset comprises approximately 300 action interactions.

UT-Kinect Action3D [43]: This dataset collects data at a fixed frame rate of 15 fps using a fixed Kinect and Kinect for Windows SDK Beta version depth camera, including RGB, depth, and 3D skeleton data. UT-Kinect divides the sample into 10 daily life behaviors, including walking, sitting, standing up, picking up, carrying, throwing, pushing, pulling, waving, and clapping. These actions are performed by 10 different individuals, with each person performing the same action twice, resulting in a total of 199 action sequences.

Florence 3D [44]: This dataset collects data through a fixed Kinect and collects nine common indoor action categories, such as “watching”, “drinking water”, and “calling.” In these actions, 10 people completed 9 actions, repeating each action 2 or 3 times, for a total of 215 actions.

TGAD: At present, the publicly available dataset of gesture actions does not include specific teaching gesture behaviors. Therefore, to assist in recognizing teaching gesture actions in teaching scenarios, this paper constructs a dataset called TGAD which shown in Figure 5. This dataset contains four types of teaching gesture actions (i.e., casual, indicative, descriptive, and operational gestures), totaling 400 skeletal action sequences. These gesture movements are derived from classroom teaching videos in primary and secondary schools from various perspectives.

### 4.2. Evaluation Metrics

In action recognition, the accuracy (AC) of behavior recognition is commonly used as the evaluation indicator for various methods, which is defined as Equation (Equation 10):(10)$$AC=\frac{N_{\mathrm{correct}}}{N_{\mathrm{sum}}}.$$

Among them, $N_{\mathrm{correct}}$ represents the number of correctly classified samples; $N_{\mathrm{sum}}$ represents the total number of samples. In the kinetics dataset, two evaluation methods are used: top-1 (the probability that the category with the highest predicted score is the same as the actual category) and top-5 (the probability that the top five predicted categories contain the actual category). All other datasets were evaluated using top-1.

Cross-Subject: This refers to the C-Sub protocol, which is a standard for dividing training and testing sets in the NTU RGB+D 60 and NTU RGB+D 120 datasets. In NTU RGB+D 60, C-Sub selects 20 people with different character numbers as the training set, and the remaining as the test set. In NTU RGB+D 120, C-Sub divided 106 participants equally into a training group and a testing group.

Cross-View: This belongs to the classification standard for training and testing sets in the NTU RGB+D 60 dataset, abbreviated as the C-View protocol. The C-View standard divides the training and testing sets by camera. The samples collected by cameras 1 and 2 are used as the test set, while the samples collected by camera 3 are used as the training set.

### 4.3. Implementation Details

We implemented MoGRU using the PyTorch framework. The data input uses a raw sequence of human skeleton keypoint coordinates. The data in the dataset used in this article will be preprocessed as N×K, where *N* represents the time frame of each action and *K* represents the coordinate information of each joint point in 2D or 3D form. During the processing of the training set, the input data of the network are normalized by z-scores. We use an SGD solver and an initial learning rate of 0.001 to train our model. All experiments have a mini-batch size of 64 and a training period of 50, except for experiments on NTU RGB+D where the mini-batch size is 256 and the training period is 150. The training was conducted on a machine equipped with two Nvidia GeForce RTX 2080 GPUs (Santa Clara, CA, USA), an Intel Core i9-9900K CPU processor (Santa Clara, CA, USA) with 32 cores, and 63.9 GB of RAM. Unless otherwise specified, both GPUs are used to allocate mini-batch training between two cards.

### 4.4. Results and Analysis

To better test the recognition performance of the MoGRU network on teaching gesture actions, we randomly partitioned the TGAD dataset into five parts, selected four parts as the training set of the model through fivefold cross-validation, and used the remaining part as the validation set. The recognition accuracy of the model on the validation set was assessed to evaluate its recognition ability. Figure 6 and Figure 7 display the recognition results of the MoGRU action recognition network with fivefold cross-validation for various gesture actions. The confusion matrix presents the predicted and actual results of each label recognition. Among them, the number of four gesture categories tested in each round is 80, with the highest recognition accuracy being operational gestures and the lowest being indicative gestures. The final teaching gesture recognition accuracy of the model is calculated to be 93.5%. This result also proves that the MoGRU model proposed in this paper has excellent performance in classifying teaching gesture actions. At the same time, to verify the reliability of the TGAD dataset production and the powerful performance of the MoGRU model for teaching gesture action recognition, we also attempted to compare the recognition capabilities of some publicly available benchmark models on TGAD. The ratio of the training and testing sets was 8:2, and the training period was 50 epochs. In the training process, the learning rate used is 0.0001 and the data augmentation algorithm is used to avoid overfitting problems. Table 1 shows the recognition accuracy of the proposed MoGRU model and benchmark model on TGAD.

### 4.5. Baseline

To demonstrate the superior performance of the MoGRU model, we compared it with existing baseline models on publicly available benchmark action recognition datasets.

**Multi-task DL** [50]: This article uses a multi-task deep learning approach for action recognition and 2D/3D human pose estimation.

**Glimpse Clouds** [51]: This method does not directly rely on learning the posture information of the human body but predicts information related to action classification through the visual attention mechanism module.

**ST-GCN** [20]: This article proposes a novel model called ST-GCN, which is used to handle dynamic skeletal architectures and compensates for the shortcomings of most previous methods by automatically learning spatial and temporal patterns.

**CoAGCN** [52]: This method constructs an efficient skeleton-based online action recognition method by stepwise inputting continuous frame sequences into a graph convolutional network.

**3s-ActCLR** [53]: This article proposes an action-dependent contrastive learning method to achieve adaptive modeling of motion and static separately.

**Sem-GCN** [54]: To address the problem of limited representation in skeleton feature maps, this paper proposes a new semantic guided graph convolutional network.

**3s RA-GCN** [55]: To avoid interference from incomplete joint information, this paper proposes a multi-stream network model based on GCN structure, which enhances the robustness of the model through branch information fusion.

**PGCN-TCA** [56]: A pseudo-GCN model with time and channel attention to observe feature information between disconnected joint points.

**Hands Attention** [57]: This article proposes a dual-stream network model for integrating pose information and RGB information, which fully utilizes the behavioral features in video data.

**Lie Group** [46]: Unlike previous methods of using joint positions or angles to represent the human skeleton, this paper proposes a special spatial representation method that uses 3D geometric relationships.

**ST LSTM + Trust Gates** [58]: This article proposes a tree structure based on RNN structure for learning representation, and adds a gating unit to LSTM to explore the spatio-temporal characteristics in skeletal action sequences.

**SCK + DCK** [47]: By exploring the spatio-temporal characteristics of skeleton sequences and the vector representation of dynamic information, the accuracy of action recognition is enhanced.

**LSTM + FA + VF** [59]: This article observes the intrinsic characteristics of action sequences from multiple perspectives and then fuses information based on different LSTM structures.

**Elastic Functional Coding** [60]: This method proposes an elastic function encoding method based on human behavior by studying from vector fields to latent variables, which can be used for human action recognition.

**Relative 3D geometry** [61]: This article proposes a new skeleton representation method R3DG, which explicitly reconstructs and expresses human joint parts in 3D.

**VA-LSTM** [48]: To mitigate the impact of action view changes, this article designs a new view adaptation scheme that automatically determines the virtual observation viewpoint during the action process through data-driven methods.

**Temporal Subspace Clustering** [62]: This article improves the previous problem of pruning skeleton information sequences through unsupervised learning, mainly proposing a space clustering method.

### 4.6. Comparison with Baseline Methods

To validate the robustness of the action recognition network MoGRU model, we also trained and tested it on the following publicly available benchmark action recognition datasets: NTU RGB+D 60, UT-Kinect Action3D, SBU Kinect Interaction, and Florence 3D. We believe that these datasets cover a wide range of real-world interactions, with varying numbers of participants, viewpoint changes, and input devices, and it is sufficient to verify the robustness and strong generalization ability of the model proposed in this article

First, on the publicly available large action dataset NTU RGB+D, we divide the dataset into two types based on different evaluation metrics C-Sub and C-View. We trained and tested the MoGRU action recognition network separately under different data partitioning standards. The training set accounts for 80% and the testing set accounts for 20%. Moreover, according to the author’s note on the dataset, we did not use 302 missing or incomplete skeleton data samples during training and testing. For each action data sample’s 3D skeleton data information, we uniformly convert it into a 75-dimensional vector form in the order of the action sequence (a total of 15 human keypoints). The results indicate that although the MoGRU action recognition network model proposed in this article only uses the raw skeleton information, it can still achieve similar testing results to other existing action recognition models. Table 2 presents the recognition accuracy of the proposed model under two standards.

Second, to verify the excellent performance of the MoGRU action recognition network in dealing with insufficient data, this paper trained and tested it on some small datasets such as SBU Kinect Interaction, Florence 3D, and UT-Kinect Action3D. For these, the ratio of the training and testing sets was 8:2. We also used dropout (0.5) and data augmentation to avoid overfitting. In terms of data sample processing, we referred to vector conversion methods similar to those mentioned above for the NTU RGB+D dataset. The experimental results show that compared with traditional deep models that construct deep network structures and train a large amount of data, the MoGRU action recognition network proposed in this paper can achieve excellent performance on numerous small datasets with a simple and efficient model structure. Table 3, Table 4 and Table 5 present the identification accuracy based on the *A* evaluation metrics for each dataset.

Compared with most existing baseline models, the MoGRU network model proposed in this paper exhibits superior action recognition accuracy. At the same time, when comparing the SOTA models in action recognition on some datasets, the model proposed in this paper can also maintain similar recognition accuracy in a lightweight structure without any preprocessing, such as using additional image information (RGB or depth) or other datasets for pretraining. This also ensures real-time recognition of teaching gesture actions. Finally, these test results further illustrate that the proposed model has strong generalization ability and can complete recognition of actions beyond teaching gestures.

### 4.7. Ablation Experiments

#### 4.7.1. Comparative Experiments of BiGRU

To demonstrate the advantages of the BiGRU compared with LSTM structures, we conducted ablation experiments on the recognition rate and accuracy of the model. The test dataset used is SBU Kinect Interaction, with a training cycle of 60 epochs and a batch size of 64. Table 6 displays the specific results of the experiment. In terms of recognition rate, the GRU network structure can demonstrate superior performance compared with the LSTM network structure. In terms of recognition accuracy, the GRU structure can ensure similar recognition effects compared with the LSTM network structure. Moreover, if a bidirectional connection strategy is adopted, the recognition accuracy of the model can be further improved. Therefore, in practical applications, to ensure sufficient recognition accuracy and achieve a faster recognition speed, this model adopts a stacked three-layer BiGRU network structure as the backbone of the model.

#### 4.7.2. Comparative Experiments of Co-Attention

To investigate the impact of the improved attention mechanism module on the overall performance of the model, we assessed the recognition accuracy using different attention mechanism models on the UT-Kinect Action3D dataset. The training process adopts a training cycle of 60 epochs and a batch size of 32. Table 7 demonstrates that using the attention mechanism module can better capture sequence features, while using the improved attention mechanism module yields superior recognition accuracy than using the general self-attention mechanism module and multi-head attention mechanism module. This verifies that the attention mechanism module proposed in this article can better integrate the spatio-temporal features of action sequences.

#### 4.7.3. Comparative Experiments of RTMPose

To illustrate the impact of human keypoint detection models on subsequent action recognition and classification, this paper presents the detection performance of the RTMPose model used in this paper and the existing baseline model (HRNet [29]) in the form of images. The results indicate that in real classroom teaching scenarios, the problem of self-occlusion by teachers and occlusion between objects or people is difficult to avoid. Using a precise and efficient human pose detector is beneficial for alleviating this problem, thereby bringing more accurate classification results for subsequent gesture recognition. Figure 8 shows the specific results of the detection.

## 5. Conclusions

This paper introduces a teaching gesture recognition algorithm based on skeletal keypoints to reduce the the difficulty of single-target dynamic gesture recognition in multi-person teaching scenarios. This algorithm employs human pose estimation technology to extract skeletal keypoint information of teachers in classroom videos and then segments the extracted bone sequence into gesture actions through action recognition technology. The experimental results demonstrated that this algorithm can accurately partition teaching gesture actions in a short period. To validate the generalization of the action recognition network, we also conducted tests and evaluations on different benchmark action datasets. The results indicate that compared with most existing SOTA models, the action recognition network constructed in this paper exhibits superior performance. We also conducted various ablation experiments on the model structure of the network to illustrate the feasibility and effectiveness of the network module design.

In future work, we will persist in refining the categories of teaching gestures for teachers and strive to be more diverse and comprehensive. In terms of model design, we will also endeavor to integrate some network modules that are more sensitive to target spatio-temporal information to enhance the model’s discriminative ability. In addition, considering the integration of more data information (such as RGB information and depth information) in action recognition processing, achieving multi-modal processing of data will become an option.

## Figures and Tables

**Figure 1 sensors-24-02589-f001:**
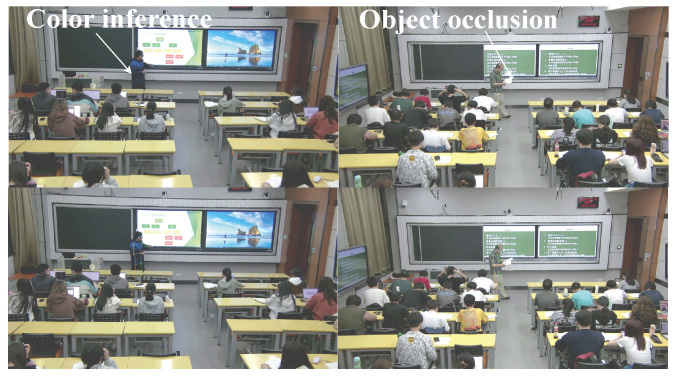
Difficulties in extracting human keypoints and the detection result of the pose estimation model used in this paper.

**Figure 2 sensors-24-02589-f002:**
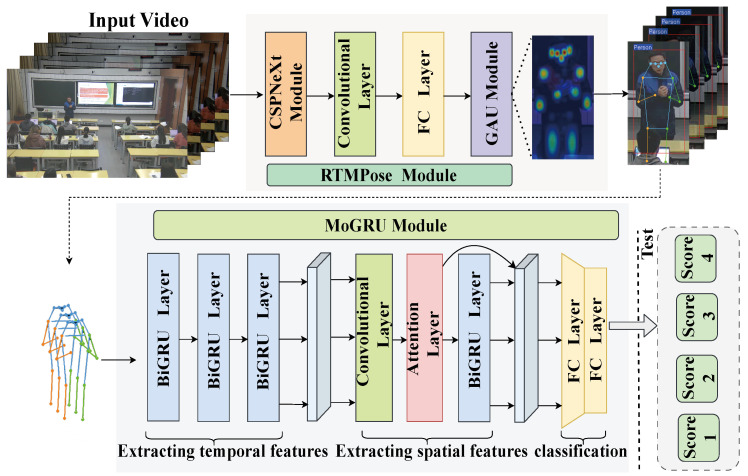
Overall architecture of the ST-TGR network model. The RTMPose module is responsible for extracting keypoint information of teachers’ skeletons from video frame images, while the MoGRU module performs gesture classification on the extracted skeleton sequence.

**Figure 3 sensors-24-02589-f003:**
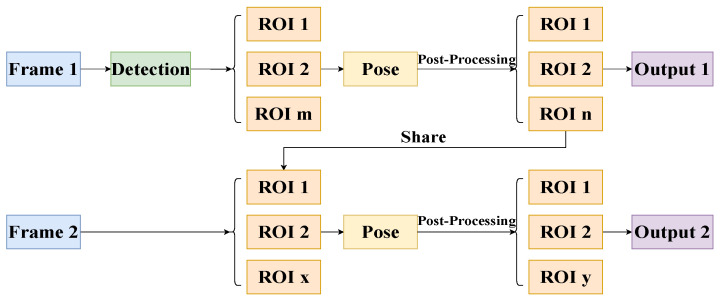
Inference pipeline of pose estimation. To accelerate the inference speed of the model, frame skipping detection was adopted in RTMPose.

**Figure 4 sensors-24-02589-f004:**
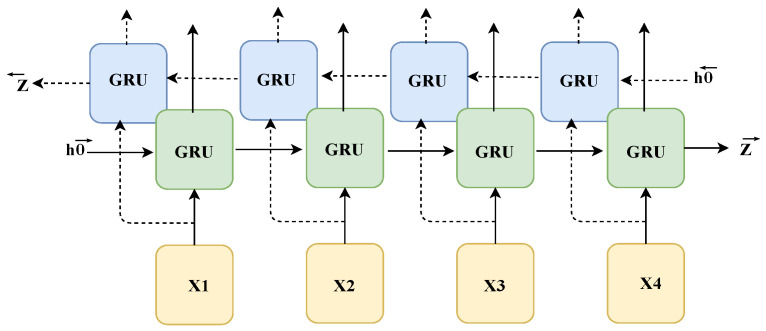
Structure of BiGRU. It consists of two independent GRU layers, one processing the sequence in the forward direction and the other processing the sequence in the reverse direction.

**Figure 5 sensors-24-02589-f005:**
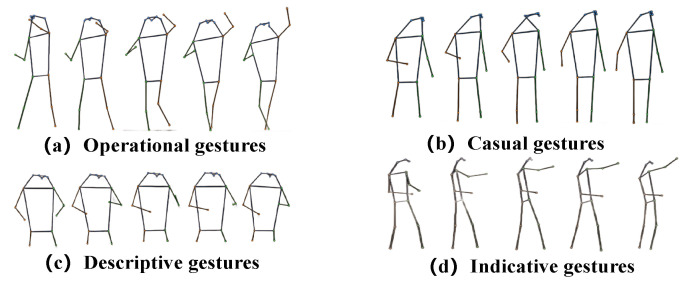
Visualization of TGAD dataset samples. (**a**) Writing on the blackboard. (**b**) Without obvious intention behavior. (**c**) Describing the teaching content. (**d**) Pointing to the teaching content.

**Figure 6 sensors-24-02589-f006:**
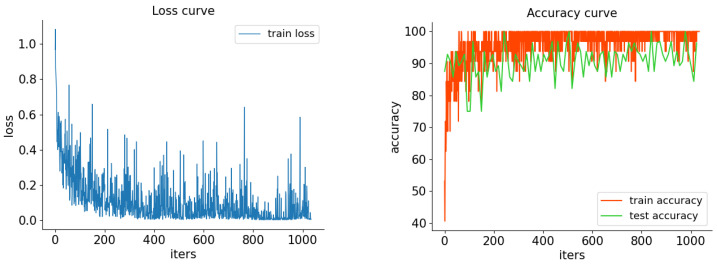
Results on TGAD dataset. (**left**) The loss value of training iterations. (**right**) The recognition accuracy of training and testing.

**Figure 7 sensors-24-02589-f007:**
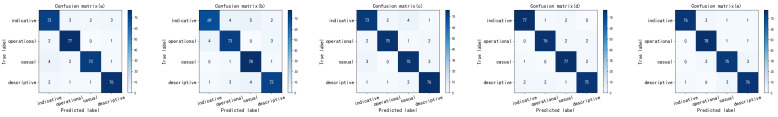
Fivefold cross-validation evaluation results on TGAD. The confusion matrix presents the difference between predicted labels and actual labels.

**Figure 8 sensors-24-02589-f008:**
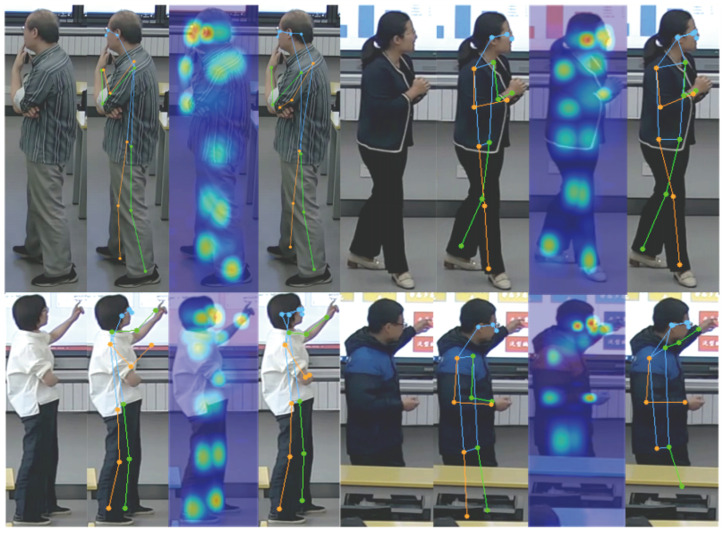
Displayed from left to right are the original image, HRNet detection image, RTMPose heatmap detection image, and RTMPose detection image.

**Table 1 sensors-24-02589-t001:** Results on TGAD dataset.

Method	Accuracy
Deep LSTM [41]	85.4
TCN [45]	86.7
Lie Group [46]	88.3
SCK + DCK [47]	89.2
VA-LSTM [48]	90.5
DeepGRU [49]	91.2
MoGRU	**93.5**

**Table 2 sensors-24-02589-t002:** Results of two evaluation metrics on NTU RGB+D 60 dataset.

Modality	Method	Accuracy	
		C-Sub	C-View
Image	Multi-task DL [50]	84.6	-
	Glimpse Clouds [51]	86.6	93.2
Pose + Image	Hands Attention [57]	84.8	90.6
	Multi-task DL [50]	85.5	-
Pose	VA-LSTM [48]	79.4	87.6
	ST-GCN [20]	86.0	93.4
	CoAGCN [52]	84.1	92.6
	3s-ActCLR [53]	84.3	88.8
	CoAGCN (2-stream) [52]	86.0	93.1
	Sem-GCN [54]	86.2	94.2
	CoS-TR [52]	86.3	92.4
	CoST-GCN [52]	86.3	93.8
	3s RA-GCN [55]	87.3	93.6
	PGCN-TCA [56]	88.0	93.6
	MoGRU	**88.5**	**93.7**

**Table 3 sensors-24-02589-t003:** Results on SBU Kinect Interaction dataset.

Modality	Method	Accuracy
Image	Hands Attention [57]	72.0
Pose + Image	Hands Attention [57]	94.1
Pose	Hands Attention [57]	90.5
	ST LSTM + Trust Gates [58]	93.3
	GCA-LSTM [63]	94.1
	LSTM + FA + VF [59]	95.0
	VA-LSTM [48]	**97.2**
	MoGRU	96.3

**Table 4 sensors-24-02589-t004:** Results on UT-Kinect Action3D dataset.

Method	Accuracy
ST LSTM + Trust Gates [58]	97.0
Lie Group [46]	97.1
SCK + DCK [47]	98.2
GCA-LSTM [63]	98.5
Temporal Subspace Clustering [62]	99.5
MoGRU	**99.7**

**Table 5 sensors-24-02589-t005:** Results on Florence 3D Action dataset.

Method	Accuracy
Elastic Functional Coding [60]	89.6
Relative 3D geometry [61]	90.7
Lie Group [46]	90.9
SCK + DCK [47]	95.2
Temporal Subspace Clustering [62]	95.8
MoGRU	**96.3**

**Table 6 sensors-24-02589-t006:** Results of testing different recurrent structures on SBU Kinect Interaction dataset.

Unit	Stacked	Time (s)	Accuracy
LSTM	3	213	94.2
LSTM	5	453	94.9
BiLSTM	3	549	95.6
BiLSTM	5	627	96.7
GRU	3	145	93.8
GRU	5	399	94.4
**BiGRU**	**3**	**517**	**96.4**
BiGRU	5	564	96.0

**Table 7 sensors-24-02589-t007:** Results of testing different attention mechanism modules on UT-Kinect Action3D dataset.

Module	Accuracy
Fully Connected (Not Using Attn)	90.2
Self-Attention	96.6
Multi-Head Attention	97.5
**Co-attention**	**99.7**

## Data Availability

The raw data supporting the conclusions of this article will be made available by the authors on request.

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
