# Peer review of "ST-TGR: Spatio-Temporal Representation Learning for Skeleton-Based Teaching Gesture Recognition"

_sensors, 2024, doi:10.3390/s24082589_

Round 1

Reviewer 1 Report

Comments and Suggestions for Authors

The article presents a method for teaching gesture recognition using skeletal keypoints extracted from classroom videos. It addresses the challenge of recognizing dynamic gestures in multi-person teaching scenarios. The proposed method integrates human pose estimation and action recognition technologies to accurately partition teaching gestures. The study includes experiments on benchmark action corpora and various experiments to validate the effectiveness of the proposed model.

Positive aspects:

1) The use of skeletal keypoints to teach gesture recognition.

2) The inclusion of experiments on benchmark datasets and ablation experiments.

3) The results indicate that the proposed model demonstrates its effectiveness.

Negative aspect:

1) The section discussing related methods is conspicuously lacking in depth and breadth. Recent advances in gesture recognition, especially those achieved on widely recognized benchmarks such as the AUTSL corpus, are conspicuously absent from the discussion. The failure to include references to state-of-the-art approaches that are readily available on platforms such as Papers with Code (https://paperswithcode.com/sota/sign-language-recognition-on-autsl) undermines the relevance of the research. For example, recent top-performing methods on the AUTSL corpus, such as STF+LSTM, SAM-SLR, and FE+LSTM, have demonstrated remarkable accuracy. These methods use sophisticated techniques such as spatio-temporal fusion (STF), self-attention mechanisms (SAM), and feature extraction with LSTM networks (FE+LSTM) to achieve performance in gesture recognition tasks.

2) The article provides only a cursory overview of the proposed method and does not provide sufficient detail for replication or comprehensive understanding. Critical aspects such as data preprocessing steps, model architecture specifics, and training procedures are inadequately described, leaving the researcher uncertain about the reproducibility and reliability of the results.

3) Although the language used in the article is generally understandable, it suffers from a lack of clarity and precision. Complex technical concepts are poorly explained. This lack of linguistic clarity significantly hampers the ability to discern the novel insights presented in the research.

I recommend a major revision of the article in its current form.

Comments on the Quality of English Language

Moderate editing of English language required.

Author Response

Thank you for your valuable feedback on our work. In response to your suggestions, we have made the following response:

1. We have added some relevant methods on the AUTSL dataset, such as STF+LSTM and SAM-SLR as you mentioned. The method proposed in this article is an action recognition technique based on skeleton key points, which can be widely applied to various action recognition tasks (not just sign language recognition). In comparative experiments, the action datasets of different sizes used in this article can further prove this viewpoint.

2. Provide as detailed as possible the methods proposed in this article and some details adopted in the experimental process, including the dimensional information of the input data and processed feature vectors of the proposed model.

3. Adjusted the description of certain technical concepts to better understand the content of this article.

The above are the modifications made in response to your feedback. Thank you again for your suggestions. Please see the attachment for detailed modifications.

Reviewer 2 Report

Comments and Suggestions for Authors

1.Please cite the source of RTMPose.

2.It is suggested to detail the dimension or shape of the input videos, skeletons and features.

3.It is suggested to detail how the attention layer works in the form of formulas or diagrams.

4.The plots in Figure 7 are not clear and need to be enlarged.

5.Since this article focuses on teaching gesture recognition, it is suggested to compare the MoGRU model with baseline models on the TGAD dataset.

Comments on the Quality of English Language

Basically readable, with some grammatical issues that need to be corrected.

Author Response

Thank you for your valuable suggestions on our work. In response to your suggestions, we have made the following response:

1. Add reference to RTMPose (line 196)

2. Provide as detailed a description as possible of the input video, skeleton, and features (lines 247, 263, 281, 295)

3. Described the operation of attention mechanism in the form of a formula (300 lines)

4. Resize Figure 7 (Page 11)

5. Added a comparison experimental tables between benchmark models and MoGRU proposed in this article on the TGAD dataset (Line 387)

The above are the modifications made in response to your advice. Thank you again for your suggestions. Please see the attachment for detailed modifications

Round 2

Reviewer 1 Report

Comments and Suggestions for Authors

The authors of the article responded to all comments. Each response was logically justified. Also, all deficiencies have been corrected, and the article has been expanded. In this form, the article can be recommended for publication.

Comments on the Quality of English Language

Minor editing of English language required.